# The Relationship between Under-Nutrition and Hypertension among Ellisras Children and Adolescents Aged 9 to 17 Years

**DOI:** 10.3390/ijerph17238926

**Published:** 2020-12-01

**Authors:** Tumisho Praise Mphahlele, Kotsedi Daniel Monyeki, Winnie Maletladi Dibakwane, Sekgothe Mokgoatšana

**Affiliations:** 1Department of Physiology and Environmental Health, University of Limpopo, Sovenga 0727, South Africa; misho.mphahlele@gmail.com (T.P.M.); winnie.dibakwane@ul.ac.za (W.M.D.); 2Department of Cultural and Political Studies, School of Social Sciences, University of Limpopo, Sovenga 0727, South Africa; Sekgothe.Mokgoatsana@ul.ac.za

**Keywords:** under-nutrition, African children, hypertension, muscle area, underweight

## Abstract

*Background*: Globally, under-nutrition and hypertension in children has been associated with the risk of developing cardiovascular disease (CVD) in adulthood. The main objective of this study was to investigate the relationship between under-nutrition and hypertension, furthermore, to determine the risk of developing hypertension due to under-nutrition. *Methods*: The study comprised of 1701 participants (874 boys and 827 girls) between the ages of 9 and 17 years old. All anthropometric and blood pressure measurements were taken according to standard procedures. Mid-upper arm circumference (MUAC), body mass index (BMI), upper arm fat area (UFA), total upper arm area (TUAA) and upper arm muscle area (UMA) of Ellisras children were compared with the National Health and Nutrition Examination Survey III reference population. The linear regression models were used to determine the relationship between under-nutrition with hypertension for unadjusted and then adjusted for age and gender. The logistic regression model was used to determine the risk of under-nutrition on developing hypertension for unadjusted and adjusted for age and gender. *Results*: There was a positive significant (*p* < 0.0001) association between all under-nutrition variables (MUAC, BMI, UFA, TUAA and UMA) and systolic blood pressure (SBP; beta ranges between 0.84 and 2.78), and diastolic blood pressure (DBP; beta ranges between 0.3 and 1.08 before adjusting and after adjusting for age and gender (SBP, beta ranges between 0.59 and 2.00 and DBP (beta ranges between 0.24 and 0.80. *Conclusion:* The prevalence of under-nutrition was high while the prevalence of hypertension was low in this study. The mean under-nutrition variables (BMI, UFA, UMA and MUAC) of Ellisras children were far lower compared to the NHANES III reference population. Hypertension was significantly associated with under-nutrition in this study.

## 1. Introduction

Globally, under-nutrition and hypertension in children have been associated with the risk of developing cardiovascular disease (CVDs) in adulthood [1]. It has been reported that under-nutrition is one of the major public health concerns worldwide [2,3]. Hypertension in children was defined as the occurrence of systolic blood pressure (SBP) and diastolic blood pressure (DBP) levels greater than or equal to the 95th percentile of height and sex-adjusted reference levels [4]. Under-nutrition defines as being underweight for one’s age, too short for one’s age (stunted), dangerously thin (wasted) and deficient in vitamins and minerals (micronutrient malnutrition) or denotes insufficient intake of energy and nutrients to meet an individual’s needs to maintain good health [5,6,7].

The United States Health and Nutritional Examination Survey (NHANES III) provides reference data for black Americans aged 1–74 years [8], which indicate that anthropometric values below the 5th percentile imply nutritional depletion. Gibson [9] validated the use of NHANES III reference standard to define nutritional depletion. Monyeki et al. [10] and Cameron [11] reported that low weight-for-height is a characteristic of children in the Ellisras rural area for both community children and children of farm workers. A study conducted by the US National and Nutrition Examination Survey (NHANES III) found that about 50% of the children above 10 years of age with a high blood pressure reading, developed hypertension by age 20 [12]. Sawaya et al. [2] reported that undernourished children were found to have high diastolic blood pressure. In addition, it is suggested that not only intrauterine undernutrition but also its occurrence during childhood may influence the incidence of hypertension in adulthood [13,14,15].

Under-nutrition has been diagnosed using anthropometric indicators such as body mass index (BMI), neck circumference (NC), mid-upper arm circumference (MUAC) and skinfolds thickness in population studies [16,17]. Other measurements that can be used are the total upper arm area (TUAA), upper arm fat area (UFA) and upper arm muscle area (UMA) amongst many [3]. Several cross-sectional studies have shown that anthropometric indicators (BMI, neck circumference, MUAC, UMA, TUAA, UFA and skinfolds thickness) for under-nutrition are associated with blood pressure (BP) in children from developed countries [18]. The association between under-nutrition and high blood pressure in low resource settings or underdeveloped countries in later childhood and adolescence are needed given the increased rate of morbidity and mortality worldwide emanating from under-nutrition [6,19].

Preliminary Ellisras Longitudinal Study (ELS) results reported a high prevalence of under-nutrition amongst older children (7–10 years) compared to younger children [10,20]. Furthermore, Monyeki et al. [21] reported the association between hypertension and under-nutrition amongst younger children whose birth weights were recorded while Mphekgwane et al. [22] reported no significant association between hypertension and under-nutrition in younger ELS children (age 5–15 years). The aim of this study, therefore, was to determine the association between under-nutrition and hypertension among older Ellisras children and adolescents aged 9–17 years. 

## 2. Materials and Methods

### 2.1. Geographical Area

In the north-western region of Limpopo, South Africa, lies an area called Ellisras (although later formally changed to Lephalale; the former is still widely used). The population residing in this settlement is estimated to be 50,000 in 42 settlements [23]. Many of the residents have obtained employment opportunities from the Matimba and Medupi electricity power station and Iscor coal mine. Most of the residents in Ellisras community are cattle and subsistence farmers, with only a few being civil servants and teachers [24].

### 2.2. Study Population and Sampling

Details of the Ellisras Longitudinal Study (ELS) research design and sampling have been reported elsewhere [10,25]. This cross-sectional study comprised of 1701 ELS participants (874 boys and 827 girls) between the ages of 9–17 years old who were measured in November 2003.

### 2.3. Ethical Clearance

Ethics approval was obtained from the University of Limpopo’s research ethics committee (EC20000/751-768) (sic). Parents and guardians were provided with and signed informed consent.

### 2.4. Anthropometric Measurements

All the participants underwent a series of anthropometric measurements of height, weight, arm girth and triceps skinfolds according to the procedures recommended by the International Society for the Advancement of Kinanthropometry (ISAK) [26]. A Martin anthropometer was used to measure stature to the nearest 0.1 cm, and a Schoenle electronic scale to measure body mass to the nearest 0.1 kg. A steel tape was used to measure MUAC to the nearest 0.1 cm. A slim guide skinfold caliper with inter-jaw pressure of 10 g/mm^2^ surface jaw face area was used for triceps skinfold measurements taken to the nearest 0.1 mm. The BMI was defined as the weight (kg)/height squared (m)^2^. The equations of the mid-arm muscle area were drawn from the equations of Frisancho [8]:Total upper arm area (TUAA) = C^2^/4 × π);C is the MUAC;Upper arm muscle area (UMA) = [C−(T_s_ × π)]^2^/(4 × π);C is MUAC and T_s_ is the triceps skinfold;Upper arm fat area (UFA) = TTUA−UMA.

### 2.5. Blood Pressure (BP)

Using an electronic Micronta monitoring instrument (Omron), a minimum of three blood pressure (BP) readings of systolic blood pressure (SBP) and diastolic blood pressure (DBP) were taken at an interval of five minutes apart after the subject had been seated for 5 min or longer [4,27]. The bladder of the Omron device contains an electronic infrasonic transducer that monitors the BP and pulse rate, displaying these concurrently on the screen. This versatile instrument has been designed for research and clinical purposes. In a pilot study, conducted before the survey, a high correlation (r = 0.93) was found between the readings taken with the automated device and those taken with a conventional mercury sphygmomanometer [28].

### 2.6. Quality Control

All training was conducted according to the standard procedures of the International Society for the Advancement of Kinanthropometry. Reliability and validity of anthropometric measurements were reported elsewhere [25]. In summary, the absolute and relative values for intra-tester technical error of measurements (%TEM) for weight ranged from 0.12 (0.15%) to 0.31 kg (0.36%); 0.22 (0.12%) to 0.43 cm (0.32%) for height; 0.1 to 5 mm (0.3%–0.64%) for skinfolds and 0 to 2.8 cm (0%–3%) for MUAC; while the absolute and relative values for the inter-tester technical error of measurements (% TEM) for stature ranged from 0.3 to 4.16 cm (0.1%–5.01%); for body mass from 0.12 to 0.2 kg (0%–0.3%), for skinfolds from 0.2 to 6 mm (0.4%–6.8%) and MUAC from 0 to 3.4 cm (0%–4%). The %TEM was within the 5.1% acceptable rates, as reported by Norton and Olds [26].

### 2.7. Statistical Analysis

Descriptive statistics were obtained by gender for age, MUAC, weight, height, triceps skinfold and BP. MUAC, BMI, UFA and UMA of Ellisras children were compared with the National Health and Nutrition Examination Survey (NHANES) III reference population of [8]. Nembidzane [29] reported age at peak height velocity ELS children to be occurring at the age of 14.45 years for boys at 11.82 years for girls. The current sample was then divided into three groups (namely, age 9–11 years, 12–14 years and 15–17 years). The Student’s *t*-test was used to test for the significant difference for MUAC, BMI, TUAA, UFA, UMA and BP between genders.

Hypertension was defined as the occurrence of SBP and DBP levels greater than or equal to the 95th percentile of height and sex-adjusted reference levels [4,27]. The international cut-off points for under-nutrition (grade one, two and three) by gender for exact ages defined to pass through BMI of 16, 17 and 18 kg/m^2^ were used [30,31]. Under-nutrition was further determined as the score of TUAA, UFA and UMA below the 5th percentile [8]. A chi-squared test was used to compare sets of nominal data that had larger frequency counts while the Fisher’s exact test was used when frequency cells were small (less than five or ten) between genders [32].

The linear regression models were used to determine the relationship between under-nutrition with hypertension by a crude analysis (unadjusted) and then adjusted for age and gender. The logistic regression model was used to determine the risk of under-nutrition on developing hypertension while unadjusted and adjusted for age and gender. All the statistical analysis was carried out using Statistical Package for Social Science (SPSS) version 25 [33]. The statistical significance was set at *p* < 0.05.

## 3. Results

Figure 1 and Figure 2 presents mean MUAC and BMI for Ellisras children and the NHANES III reference population aged 9 to 17 years. Mean MUAC and BMI of Ellisras children was lower compared to the NHANES III reference population throughout the age range Ellisras girls showed higher mean MUAC and BMI compared to boys though out the age range.

Figure 3 and Figure 4 showed the mean UMA and UFA of Ellisras children compared to the NHANES reference population aged 9–17 years. Ellisras children showed a lower mean UMA compared to the NHANES III reference population throughout the age range while Ellisras children did not show any mean difference between the sexes through the age of 9–17 years. Mean UFA for the NHANES reference population of girls was higher than both Ellisras children throughout the age range. Ellisras girls showed a high mean UFA compared to Ellisras boys but showed similar mean values with NHANES III reference of boys population between the age of 16 and 17 years.

Table 1 shows descriptive statistics for under-nutrition variables and BP for Ellisras rural children age 9–17 years. Girls showed a significantly (*p* < 0.05–0.001) high mean height, weight, triceps, arm girth, BMI, TUAA, UMA and UFA and BP between the 12 and 14 years age group. All the under-nutrition variables showed significantly (*p* < 0.05–0.001) high mean values for girls compared to boys in the older age group (15–17 years) except for mean height and mean UMA. In the younger age group (9–11 years) Ellisras girls showed a significantly (*p* < 0.05–0.001) high mean of all the under-nutrition variables except for height and BP.

The prevalence of total under-nutrition (mild, moderate and severe) based on BMI was significantly (*p* < 0.05) high for boys compared to girls across the age categories (age category 9–11 years (56.1% vs. 49.7%), age category 12–14 years (51.6% vs. 40.8%) and age category 15–17 years (53.7% vs. 40.8%)). The prevalence of mild under-nutrition was significantly higher for boys compared to girls at the younger age categories (aged 9–11 years (38.4% vs. 30.6%) and age 12–14 years (32.4% vs. 23.2%)), while girls showed a high prevalence of under-nutrition at an older age group (age category 15–17 years (13.5% vs. 25.3%). Hypertension ranged between 2% and 5.5% across the age range with an insignificant difference between the genders.

Table 2 shows linear regression for the association between blood pressure and under-nutrition variables of Ellisras children aged 9–17 years. There was a positive significant (*p* < 0.0001) association between all the under-nutrition variables (MUAC, TUAA, UFA, UMA and BMI) and SBP (beta ranges between 0.84 (95% CI 0.76–0.91) and 2.78 (95% CI 2.54–3.02)) and DBP (beta ranges between 0.33 (95% CI 0.27–0.39) and 1.08 (95% CI 0.90–1.26)) before adjusting. Furthermore after adjusting for age and gender SBP (beta ranges between 0.59 (95% CI 0.76–0.50), 1.95 and 2.00 (95% CI 1.71–2.28)) and DBP (beta ranges between 0.24 (95% CI 0.18–0.31) and 0.80 (95% CI 0.57–1.03)).

Table 3 exhibits logistic regression (odds ratio, 95% CI and *p*-value)) for the association between elevated BP and under-nutrition variables for Ellisras children aged 9–17 years. There was a significant (*p* < 0.05) association between elevated SBP (odds ratio ranges from 0.12 (95% CI 0.02–0.84 for UFA) to 0.43 (95% CI 0.22–0.87 of moderate under-nutrition)) of all under-nutrition variables even after adjusting for age and gender (odds ratio ranged from 0.11 (95% CI 0.02–0.83 of UFA) to 0.64 (95% CI 0.42–0.97 to mild under-nutrition)). Total under-nutrition based on BMI was significantly associated (*p* < 0.05) with high diastolic blood pressure (OR = 0.60 (95% CI 0.42–0.87)) even after adjusting for age and gender (OR = 0.63 (95% CI 0.44–0.91)). Hypertension was significantly associated with total under-nutrition (mild, moderate and severe) based on BMI (OR = 0.39 (95% CI 0.21–0.73)) even after adjusting for age and gender (OR = 0.41 (95% CI 0.22–0.77)).

## 4. Discussion

This study aimed to determine the relationship between under-nutrition and hypertension in the rural area of Ellisras for children and adolescents aged between 9 and 17 years. There was a significant (*p* < 0.05) association between hypertension and total under-nutrition (mild, moderate and severe added together) based on BMI but not based on MUAC, TUAA, UFA and UMA. Silva et al. [34] reported no significant association between under-nutrition and hypertension amongst Brazilian children. In contrary, Ma et al. [35] and Mazıcıoğlu et al. [36] reported a significant association between hypertension and MUAC in Chinese and Turkish children. Furthermore, in this study, there was a significant (*p* < 0.05) association between nutritional status variables (BMI MUAC, TUAA, UFA and UMA) and both SBP and DBP. Sawaya et al. [2] only found a significant association between TUAA and high SBP amongst Sardinian children while Silva et al. [34] reported a significant association between TUAA and high SBP. 

The childhood and adolescence period are critical periods for the development of adiposity, which poses a threat later in life [37,38]. In the current study BMI, MUAC, TUAA and UMA were significantly higher in girls compared to boys in the 12–14 years age group. Similar results were reported by Piperata [39]. Jaswant and Nitish [40] reported significantly high UFA for Indian girls compared to the boys. Contrary to the present study there was no clear distinct patterns between boys and girls in mean TUAA and UFA area while boy adolescents exhibited a high mean UMA compared to girls in Bahraini adolescence [41]. Puberty adipocyte increased greatly in both girls and boys, but the girl increases far exceeded that of the boys [42].

The prevalence of under-nutrition was high in the current study. Similar trends were reported previously amongst South African children [5,6,43]. Furthermore, the mean BMI, MUAC, TUAA, UFA and UMA were far too low compared to the reference population [8], which confirms the developing nature of the African continent compared to other developed countries. Poorer people in South Africa struggled with food insecurity, lack of knowledge about the risk of chronic diseases of lifestyle at a younger age [20,44]. Consequently, the majority of these undernourished children will be at a higher risk of developing cardiovascular diseases when exposed to western diets during adulthood [44,45]. In the rural South African population, illiteracy should be the first hurdle to be jumped by health professionals, school teachers and community leaders. Furthermore, the South African population should not comply with what they have in terms of traditional knowledge and medicine, but rather seek new innovative ways of addressing issues facing the population in terms of health, diet, climate, environment and lifestyle changes in the ongoing political transformation. Healthy, active and well-nourished children are a prerequisite for sustained economic development.

The limitation of the study include that we did not consider the family history of cardiovascular diseases, which was reported to have a strong risk factor for future hypertension [46]. This study was a cross-sectional design, which prevents the conclusion of a causal association of under-nutrition and blood pressure. The possibility of bias cannot be an exception in the present study as many of the subjects might not have been aware of the risk factors for under-nutrition and chronic diseases of lifestyle given the indigenous knowledge system that is rooted in rural South African and African populations. The anthropometric and BP measurements were taken directly; hence, recall or estimation bias will not prevail in our study.

## 5. Conclusions

The prevalence of under-nutrition was high while the prevalence of hypertension was low in this study. The mean nutritional status variables of Ellisras children were far lower compared to the NHANES III reference population. There was a positive significant association between hypertension and under-nutrition based on BMI. Furthermore, studies are required to investigate the association between biochemical analyses of blood lipids profile and the lifestyle of these children to further uproot the dynamics of cardiovascular diseases in this Ellisras rural population over time.

## Figures and Tables

**Figure 1 ijerph-17-08926-f001:**
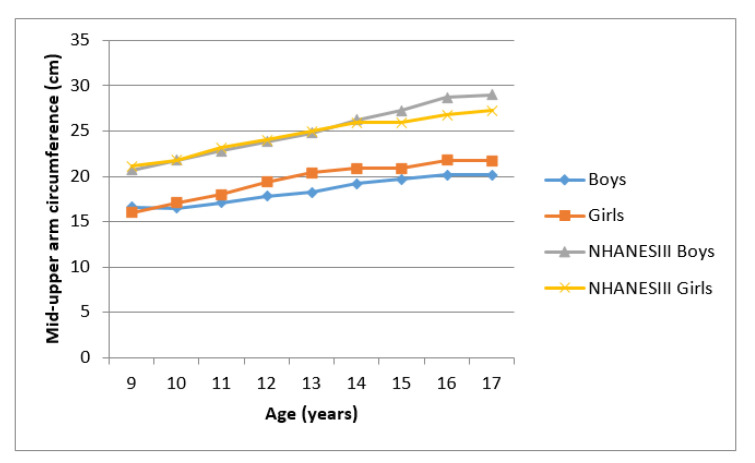
Mean mid-upper arm circumference of Ellisras children compared to the NHANES III reference population aged 9–17 years.

**Figure 2 ijerph-17-08926-f002:**
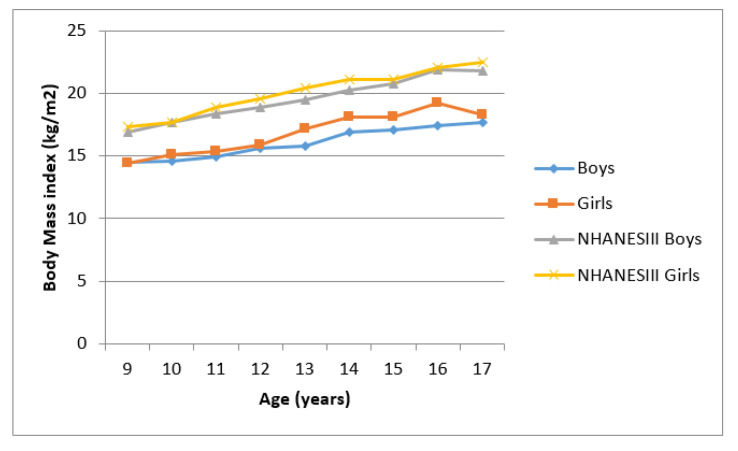
Mean body mass index of Ellisras children compared to the NHANES III reference population aged 9–17 years.

**Figure 3 ijerph-17-08926-f003:**
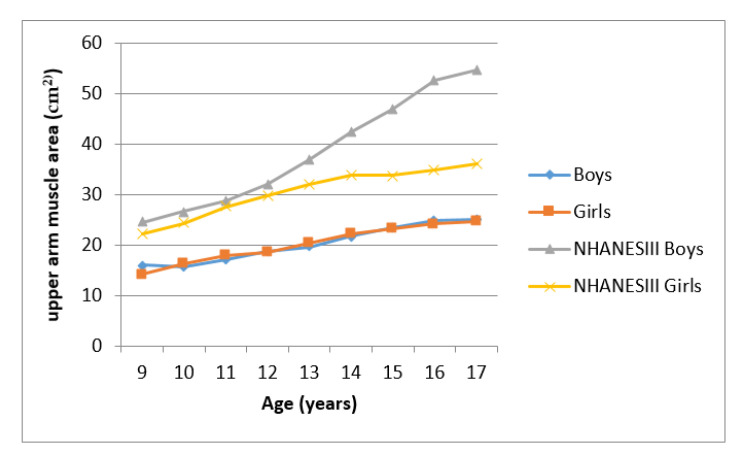
Mean upper arm muscle area of Ellisras population compared to the NHANES III reference population aged 9–17 years.

**Figure 4 ijerph-17-08926-f004:**
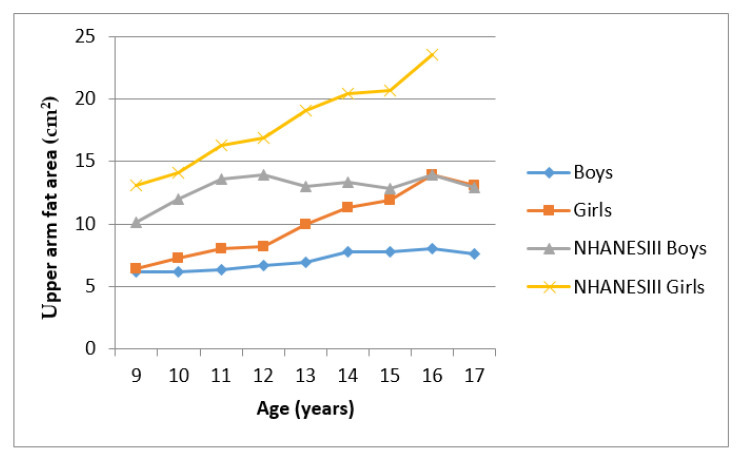
Mean upper arm fat area of Ellisras children compared to the NHANES III reference population aged 9–17 years.

**Table 1 ijerph-17-08926-t001:** Descriptive statistics for under-nutrition variables and blood pressure for Ellisras rural children age 9–17 years.

Variable	Age 9–11 Years				Age 12–14 Years				Age 15–17 Years			
	Boys		Girls		Boys		Girls		Boys		Girls	
	*N* = 164		*N* = 147		*N* = 413		*N* = 392		*N* = 296		*N* = 289	
	M	(sd)	M	(sd)	M	(sd)	M	(sd)	M	(sd)	M	(sd)
Age (years)	10.9	(0.79)	10.9	(0.72)	13.7	(0.84)	13.7	(0.82)	15.9	(0.60)	15.9	(0.62)
Arm girth (cm)	16.8 **	(1.21)	17.5 **	(1.89)	18.5 **	(1.65)	19.6 **	(2.23)	19.9 **	(2.07)	21.2 **	(2.38)
Triceps (mm)	7.9 **	(1.82)	9.4 **	(2.74)	8.3 **	(2.52)	11.2 **	(3.97)	8.4 **	(2.89)	12.9 **	(4.72)
Weight (kg)	28.7 **	(3.99)	30.4 **	(6.26)	38.1 **	(7.03)	41.9 **	(8.09)	45.6 **	(7.91)	48.4 **	(7.57)
Height (cm)	139.2	(6.80)	141	(8.24)	152.9 *	(8.55)	155.1 *	(7.55)	162.1	(9.06)	161.5	(6.25)
BMI (kg/m^2^)	14.7 **	(1.25)	15.2 **	(1.99)	16.2 *	(1.84)	17.3 *	(2.46)	17.2 *	(2.01)	18.5 *	(2.48)
TUAA (cm^2^)	22.6 *	(3.24)	24.5 *	(5.47)	27.5 **	(5.11)	30.9 **	(7.35)	31.9 **	(6.92)	36.2 **	(8.54)
UMA (cm^2^)	16.4 *	(2.49)	16.9 *	(3.91)	20.3 *	(3.61)	20.7 *	(4.03)	24.1	(5.02)	23.6	(4.53)
UAF (cm^2^)	6.19 **	(1.59)	7.6 **	(2.62)	7.2 **	(2.64)	10.2 **	(4.53)	7.9 **	(3.38)	12.6 **	(5.77)
SBP (mmHg)	94.2	(12.41)	94.1	(11.29)	99.8 *	(11.65)	102.4 *	(11.89)	107.6 *	(13.96)	111.8 *	(12.43)
DBP (mmHg)	58.2	(9.94)	58.5	(8.79)	60.5 *	(9.39)	62.1 *	(8.98)	61.5 *	(9.13)	65.6 *	(9.64)
Prevalence												
	%	(*n*)	%	(*n*)	%	(n)	%	(*n*)	%	(*n*)	%	(*n*)
TUAA	4.3	(7)	7.5	(11)	5.3	(22)	6.1	(24)	4.4	(13)	5.2	(15)
UMA	6.1	(10)	4.8	(19)	4.8	(20)	4.8	(19)	5.1	(15)	4.8	(14)
UFA	4.9	(8)	6.1	(9)	4.8	(20)	5.1	(20)	5.1	(15)	4.8	(14)
TU	56.1 *	(92)	49.7 *	(73)	51.6 *	(213)	40.8 *	(160)	53.7 *	(159)	40.8 *	(118)
Mild	38.4 *	(63)	30.6 *	(45)	32.4 *	(134)	23.2 *	(91)	13.5 *	(40)	25.3 *	(73)
Moderate	14	(23)	11.6	(17)	12.3	(51)	12.8	(50)	13.5 *	(40)	9.3 *	(27)
Severe	3.7	(6)	7.5	(11)	6.8 *	(28)	4.8 *	(19)	11.1	(33)	6.2	(18)
High systolic	3.7	(6)	4.1	(6)	6.3	(26)	7.9	(31)	12.5	(37)	15.2	(44)
HD	9.1	(15)	6.8	(10)	6.1	(25)	7.7	(30)	5.7	(17)	12.8	(37)
Hypertension	2.4	(4)	3.4	(5)	2.4	(10)	2.8	(11)	2.0	(6)	5.5	(16)

* = *p* < 0.05, ** = *p*<0.001, TUAA = Under-nutrition based on total upper arm area cut-off point; UFA = Under-nutrition based on the upper arm fat area cut-off points; UMA = Under-nutrition based on upper arm muscle area; TU = Total under-nutrition based on body mass index, M = mean, sd = standard deviation, HD = High diastolic

**Table 2 ijerph-17-08926-t002:** Linear regression (beta, *p*-value and 95% confidence interval) for the association between blood pressure and under-nutrition variables of Ellisras children aged 9–17 years.

Variables	Unadjusted				Adjusted for Age and Gender			
	β	*p*-Value	95% CI		β	*p*-Value	95% CI	
Systolic Blood Pressure								
MUAC	2.78	0.000	2.54	3.02	2.00	0.000	1.71	2.28
BMI	2.49	0.000	2.25	2.74	1.67	0.000	1.40	1.94
TUAA	0.84	0.000	0.76	0.91	0.59	0.000	0.50	1.94
UMA	1.32	0.000	1.20	1.44	0.92	0.000	0.78	1.06
UFA	1.01	0.000	0.87	1.15	0.65	0.000	0.50	0.79
Diastolic Blood Pressure								
MUAC	1.08	0.000	0.90	1.26	0.80	0.000	0.57	1.03
BMI	0.97	0.000	0.79	1.15	0.66	0.000	0.45	0.87
TUAA	0.33	0.000	0.27	0.39	0.24	0.000	0.18	0.31
UMA	0.40	0.000	0.31	0.50	0.24	0.000	0.13	0.35
UFA	0.55	0.000	0.45	0.65	0.41	0.000	0.14	1.75

MUAC = Mid-upper arm circumference, TUAA = total upper arm area; UFA = upper arm fat area; UMA = upper arm muscle area; BMI = body mass index; β = beta; CI = confidence interval.

**Table 3 ijerph-17-08926-t003:** Logistic regression with anthropometric variables as independent variables and high systolic blood pressure (SBP), high diastolic blood pressure (DBP) and hypertension as a dependent variable in the total group.

Variables	Unadjusted				Adjusted for Age and Gender			
	OR	*p*-Value	95% CI		OR	*p*-Value	95% CI	
High Systolic Blood Pressure								
TUAA	-	-	-	-	-	-	-	-
UMA	0.24	0.047	0.06	0.98	0.24	0.046	0.06	0.97
UFA	0.12	0.033	0.02	0.84	0.11	0.032	0.02	0.83
TU	0.38	0.000	0.26	0.55	0.39	0.000	0.27	0.57
Mild	0.59	0.012	0.39	0.89	0.64	0.000	0.42	0.97
Moderate	0.43	0.018	0.22	0.87	0.45	0.025	0.23	0.90
Severe	0.17	0.014	0.04	0.70	0.15	0.008	0.04	0.61
High Diastolic Blood Pressure								
TUAA	0.66	0.374	0.26	1.65	0.65	0.364	0.28	1.64
UMA	0.88	0.774	0.38	2.06	0.89	0.790	0.38	2.09
UFA	-	^-^	-	-	-	-	-	-
TU	0.60	0.007	0.42	0.87	0.63	0.015	0.44	0.91
Mild	0.72	0.125	0.48	1.09	0.76	0.169	0.50	1.15
Moderate	0.62	0.142	0.33	1.17	0.64	0.166	0.34	1.21
Severe	0.75	0.462	0.34	1.63	0.74	0.462	0.34	1.64
Hypertension								
TUAA	-	-	-	-	-	-	-	-
UMA	-	-	-	-	-	-	-	-
UFA	-	-	-	-	-	-	-	-
TU	0.39	0.003	0.21	0.73	0.41	0.006	0.22	0.77
Mild	0.51	0.065	0.24	1.05	0.54	0.096	0.26	1.11
Moderate	0.59	0.316	0.21	1.65	0.61	0.347	0.22	1.71
Severe	0.26	0.189	0.03	1.93	0.27	0.194	0.04	1.96

TUAA = Under-nutrition based on total upper arm area cut-off point; UFA = Under-nutrition based on the upper arm fat area cut-off points; UMA= Under-nutrition based on upper arm muscle area; TU = Total Under-nutrition based on BMI, OR = odds ratio, 95% Confidence interval.

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
