# Peer review of "The Relationship between Under-Nutrition and Hypertension among Ellisras Children and Adolescents Aged 9 to 17 Years"

_ijerph, 2020, doi:10.3390/ijerph17238926_

Round 1

Reviewer 1 Report

The authors investigated the relationship between under-nutrition and hypertension among children and adolescents age 9 to 17 years old, and concluded that under-nutrition and underweight are high-risk factors of hypertension.

Some issues to be underlined:

The sample must be allocated in two groups, according to the stage of development, i.e. prepubertal and pubertal, to follow the course of developmental changes. This approach has the advantage of detecting changes in pre-puberty or puberty, as well as comparing the transition from pre-puberty to puberty. (Relationship between the growth hormone/insulin-like growth factor-I axis, insulin sensitivity, and adrenal androgens in normal prepubertal and pubertal girls. Guercio G, Rivarola MA, Chaler E, Maceiras M, Belgorosky A. J Clin Endocrinol Metab. 2003 Mar;88(3):1389-93);

Endothelial function in pre-pubertal children at risk of developing cardiomyopathy: a new frontier. Tavares AC, Bocchi EA, Guimarães GV Clinics (Sao Paulo).2012;67(3):273-8.

It is necessary to add the family history of hypertension, those who have hypertensive parents have a strong risk factor for future hypertension in non-hypertensive children, regardless of other risk factors. (Effects of high-intensity aerobic interval training vs. moderate exercise on hemodynamic, metabolic and neuro-humoral abnormalities of young normotensive women at high familial risk for hypertension. Ciolac EG, Bocchi EA, Bortolotto LA, Carvalho VO, Greve JM, Guimarães GV. Hypertens Res. 2010 Aug;33(8):836-43. doi: 10.1038/hr.2010.72.

The study showed a weak correlation between subnutritional anthropometric variables and blood pressure, which means that one variable does not necessarily change the other. What is not possible to conclude that they are malnourished is likely to develop hypertension later.

Author Response

Review 1
The sample must be allocated in two groups, according to the stage of development, i.e. prepubertal and pubertal, to follow the course of developmental changes. This approach has the advantage of detecting changes in pre-puberty or puberty, as well as comparing the transition from pre-puberty to puberty. (Relationship between the growth hormone/insulin-like growth factor-I axis, insulin sensitivity, and adrenal androgens in normal prepubertal and pubertal girls. Guercio G, Rivarola MA, Chaler E, Maceiras M, Belgorosky A. J Clin Endocrinol Metab. 2003 Mar;88(3):1389-93); Endothelial function in pre-pubertal children at risk of developing cardiomyopathy: a new frontier. Tavares AC, Bocchi EA, Guimarães GV Clinics (Sao Paulo).2012;67(3):273-8.

 Response: Thank you for the comments. The changes have been effected as suggested. See  line 133, 261 and table 1.

It is necessary to add the family history of hypertension, those who have hypertensive parents have a strong risk factor for future hypertension in non-hypertensive children, regardless of other risk factors. (Effects of high-intensity aerobic interval training vs. moderate exercise on hemodynamic, metabolic and neuro-humoral abnormalities of young normotensive women at high familial risk for hypertension. Ciolac EG, Bocchi EA, Bortolotto LA, Carvalho VO, Greve JM, Guimarães GV. Hypertens Res. 2010 Aug;33(8):836-43. doi: 10.1038/hr.2010.72.

 Response:  We did not collecte the information on the history of hypertension. See the limitation of the study Line 286

The study showed a weak correlation between subnutritional anthropometric variables and blood pressure, which means that one variable does not necessarily change the other. What is not possible to conclude that they are malnourished is likely to develop hypertension later.

 Response: Thank you for the comments. we have left out the correlation tables  and hope the regression analysis could answer the question of association better as it account for confounder of age and gender. Secondly, we have let it as we wanted to keep the tables and the figures to be seven  as it is a standard peer review Journal practice. However, we have argued in the discussion that undernutrition could lead to cardiovascular diseases later in life. See line 275 to 283.

Reviewer 2 Report

The aim of this study was to determine the relationship between under-nutrition and hypertension in children and adolescents from a rural community.

General comments

The major concern with this manuscript pertains to the statistical analyses. Conclusions drawn from results are not substantiated with the analyses. The authors refer to ‘under-nutrition anthropometric’ variables in Figure 1 and Tables 3-6, however the cut-points for anthropometric variables used to identify under-nutrition are not reported anywhere and not included in all analyses. Instead, the average of the anthropometric variables is used, which includes those with normal measurements and those with measurements above average that is an indication of overnutrition or larger than the average.

Throughout language editing is required to improve grammatical errors.

Specific comments

See electronic comments on the manuscript as well as further comments below.

Abstract

Further clarification is required in the abstract. See specific comments on the electronic manuscript.

Introduction

The author refers to a causal relationship between hypertension and under-nutrition in children without sufficient evidence to support this statement. Cross-sectional study findings are referenced which is not appropriate, further evidence should be included or the statement revised.

A large body of the introduction is dedicated to describe the prevalence of childhood under-nutrition and malnutrition. It is argued that this should be shortened, and more context provided to the evidence for the association between under-nutrition and malnutrition with hypertension or cardiovascular risk in childhood. It is not clear from the introduction whether under-nutrition in children is associated with childhood hypertension or a risk factor for adulthood hypertension. Furthermore, the proposed physiological mechanisms should be briefly discussed. The current introduction does not provide sufficient background and motivation for the described study.

See specific comments in track changes on the manuscript.

Methods

Section 2.3. should include a justification for the use of TUAA, UFA and UMA as outcome anthropometric measures. Also include how many measures were taken and was an average used?

Move anthropometric calculations under section 2.4 to section 2.3.

Statistical analysis

Revisit interpretation of cut-offs for TUAA, UFA and UMA between 5th and 15th percentile.

Include reference to SPSS software.

Results

Language editing is required for the first paragraph of the results referring to Table 1. In addition, results should be presented in past tense.

Table 1 seems to have the first column missing which refers to the sample’s gender?

Table 2: Which statistical analysis were used to compare prevalence of hypertension amongst boys and girls? Include p-value. A table is not required, findings can be presented in text.

See specific comments on electronic manuscript for further comments.

The results presented in Figure 1 is not clear. The authors refer to ‘under-nutrition’, however it is not clear what the percentages of TUAA, UFA and UMA are presented in Fig 1. If under-nutrition is presented then a breakdown of the prevalence of percentiles representing wasting, average, and above average should be represented here to the reader can interpret the results themselves. Significance and p-values should be included on Figure 1 where differences were shown.

The authors refer to under-nutrition in Table 3, as per previous comment percentiles of TUAA, UFA, and UMA representing under-nutrition (i.e. <15 percentile) should be included instead of the average of these measures as they will include normal weight and overweight/above average children. In fact, the positive correlations in Table 3 suggests the higher the body mass index the higher the blood pressure which is in direct contrast to what the authors state that under-nutrition and blood pressure are correlated. UFA was negatively associated with blood pressure in all but for SBP in boys, which supports the authors statement. Again, distinguishing between the different percentile groups will be an important addition to this table.

Currently Table 3 and 4 present similar findings, it is recommended to include the suggested changes to Table 3 so that the results are distinct.

Table 5 heading: Change heading to: Logistic regression with anthropometric variables as independent variables and systolic blood pressure as dependent variable in total group, n=1701.

The independent variables in Tables 5 and 6 seem to be average anthropometric variables and not those specific to under-nutrition as per definition in the statistical analysis section. Therefore, the conclusions drawn from these findings relating to under-nutrition needs revising.

Discussion

The discussion is scant and need to include further reference to pertinent studies in the field. A larger body of the discussion merely restate the findings/results of the study.

Author Response

Review 2

The aim of this study was to determine the relationship between under-nutrition and hypertension in children and adolescents from a rural community.

General comments

The major concern with this manuscript pertains to the statistical analyses.

Response: statistical analysis section has been rewritten. See line 131 to 153

Conclusions drawn from results are not substantiated with the analyses.

Response: Thank you for the comments. The conclusion is based on the analysis of the study. See line 295 to 300.

The authors refer to ‘under-nutrition anthropometric’ variables in Figure 1 and Tables 3-6, however the cut-points for anthropometric variables used to identify under-nutrition are not reported anywhere and not included in all analyses. Instead, the average of the anthropometric variables is used, which includes those with normal measurements and those with measurements above average that is an indication of overnutrition or larger than the average.

Response: Only undernutrition data have been reported in the results section. The discussion of the study is based on the undernutrition and hypertension of the study. Furthermore, the mean values of Ellisras sample have been compared with the NHANES III reference population. See figure 1 to 4.

Throughout language editing is required to improve grammatical errors.

 Response: A qualified English native speaking professional editor assisted in editing the manuscript.

Specific comments

See electronic comments on the manuscript as well as further comments below.

Abstract

Further clarification is required in the abstract. See specific comments on the electronic manuscript.

 Response: Abstract has been corrected as suggested

Introduction

The author refers to a causal relationship between hypertension and under-nutrition in children without sufficient evidence to support this statement. Cross-sectional study findings are referenced which is not appropriate, further evidence should be included or the statement revised.

 Response: The information has been deleted. Thank you for the comments.

A large body of the introduction is dedicated to describe the prevalence of childhood under-nutrition and malnutrition. It is argued that this should be shortened, and more context provided to the evidence for the association between under-nutrition and malnutrition with hypertension or cardiovascular risk in childhood. It is not clear from the introduction whether under-nutrition in children is associated with childhood hypertension or a risk factor for adulthood hypertension. Furthermore, the proposed physiological mechanisms should be briefly discussed. The current introduction does not provide sufficient background and motivation for the described study.

 Response Thank you for the comments. The introduction has been rewritten and the comments of the reviewers has been taken into consideration in rewriting the introduction

See specific comments in track changes on the manuscript.

 Response: Changes effected as suggested

Methods

Section 2.3. should include a justification for the use of TUAA, UFA and UMA as outcome anthropometric measures. Also include how many measures were taken and was an average used?

 Response: Thank you for the comments. The quality control for the anthropometric measurements has been reported. See line120 to 229

Move anthropometric calculations under section 2.4 to section 2.3.

 Response: Anthropometric measurements has been put under anthropometric measurements as suggested by the reviewer

Statistical analysis

Revisit interpretation of cut-offs for TUAA, UFA and UMA between 5th and 15th percentile.

 Response: Only the undernutrition cut points for TUAA, UFA and UMA have been reported which is less than the 5th percentile.

Include reference to SPSS software.

 Response: The reference has been included as suggested by the reviewer.

Results

Language editing is required for the first paragraph of the results referring to Table 1. In addition, results should be presented in past tense.

 Response: Changes was effected as suggested

Table 1 seems to have the first column missing which refers to the sample’s gender?

 Response: Changes effected as suggested. Both prevalence, mean and standard deviation has been included in table 1.

Table 2: Which statistical analysis were used to compare prevalence of hypertension amongst boys and girls? Include p-value. A table is not required, findings can be presented in text.

 Response: Changes effected as suggested. See line145 “A chi-squared test was used to compare sets of nominal data that had larger frequency counts while the Fisher’s exact test was used when frequency cells were small (less than five or ten) between genders (Lauer & Clarke, 1989, Altman, 1991).”

See specific comments on electronic manuscript for further comments.

 Response: Changes effected as suggested.

The results presented in Figure 1 is not clear. The authors refer to ‘under-nutrition’, however it is not clear what the percentages of TUAA, UFA and UMA are presented in Fig 1. If under-nutrition is presented then a breakdown of the prevalence of percentiles representing wasting, average, and above average should be represented here to the reader can interpret the results themselves. Significance and p-values should be included on Figure 1 where differences were shown.

 Response: Thank you for the comments. The results are more clear. Figure 1-4 presents the mean MUAC, BMI, UFA and UMA of ELS sample compared to the NHANES III reference population. The prevalence has been included in table 1.

The authors refer to under-nutrition in Table 3, as per previous comment percentiles of TUAA, UFA, and UMA representing under-nutrition (i.e. <15 percentile) should be included instead of the average of these measures as they will include normal weight and overweight/above average children. In fact, the positive correlations in Table 3 suggests the higher the body mass index the higher the blood pressure which is in direct contrast to what the authors state that under-nutrition and blood pressure are correlated. UFA was negatively associated with blood pressure in all but for SBP in boys, which supports the authors statement. Again, distinguishing between the different percentile groups will be an important addition to this table.

 Response: Thank you for the comments. we have left out the correlation tables  and hope the regression analysis could answer the question of association better as it account for confounder of age and gender. Secondly, we have let it as we wanted to keep the tables and the figures to be seven  as it is a standard peer review Journal practice. However, we have argued in the discussion that undernutrition could lead to cardiovascular diseases later in life. See line 275 to 283.

Currently Table 3 and 4 present similar findings, it is recommended to include the suggested changes to Table 3 so that the results are distinct.

 Response: Table 3 has been left out.

Table 5 heading: Change heading to: Logistic regression with anthropometric variables as independent variables and systolic blood pressure as dependent variable in total group, n=1701.

 Response: changes effected as suggested.

The independent variables in Tables 5 and 6 seem to be average anthropometric variables and not those specific to under-nutrition as per definition in the statistical analysis section. Therefore, the conclusions drawn from these findings relating to under-nutrition needs revising.

 Response: Changes effected as suggested.

Discussion

The discussion is scant and need to include further reference to pertinent studies in the field. A larger body of the discussion merely restate the findings/results of the study.

 Response: Changes effected as suggested. Thank you for the comments.

Author Response

Reviewer 3

 Introduction

  1. Author mentioned a definition of hypertension in children. It would be better to provide a definition of under-nutrition as well. Below sentences may need to relocate after a definition of nutrition.

 Response: Definition of undernutrition has been stated as suggested.

“It can be determined using many indices and equations, namely, Body Mass Index (BMI), and waist-to-height ratio (WtHr) [7]. Other measurements that can be used are the Total Upper Arm Area (TUAA), Upper Arm Fat Area (UFA) and Upper Arm Muscle Area (UMA) amongst many [8]. “

 Response: Changes effected as suggested

  1. Reference 1 was published in 2005. Please provide more recent reference if available.

 Response: Recent reference has been cited

Parapragh2. “ there has been a causal relationship established between under-nutrition and hypertension.” This sentence seems to need reference(s).

 Response: The sentence has been deleted

  1. Page 1 last sentence “In 1998 the prevalence of Hypertension in South Africa was at 21% of the population” and Page 2. First sentence, By 2008, the prevalence had accelerated to 77.3%. Both sentences need references and please find more updated statistics.

 Response: the paragraph has been rewriter.

  1. Overall introduction is hard to read. Ideas are scattered throughout the introduction.

 Response: The introduction has been rewritten.

  1. Author mentioned “However, there was no study published about the association between under-nutrition and hypertension among rural children and adolescents aged 9 to 17 years. The aim of this study, therefore,…..” Please provide reasons why authors chose to study rural area of selected population.

 Response: The changes effected as suggested. See the introduction section

Materials and methods

  1. IRB approval number should be provided.

 Response: The approval number has been included. See line 88

  1. Need to elaborate why this chosen area is worth studying. For example, comparisons between overall populations’ demographic characteristics and this chosen area populations’ demographic characteristics should be reported.

 Response: Changes effected as suggested.

  1. Statistical analysis: 6th sentence “Measurement …………pressure” Equal to 90th is missing in the sentence. Please revise 2nd and 3rd paragraphs of the statistical analysis.

 Response: Changes effected as suggested. See the statistical analysis section line 131 to 153

  1. Please pay attention to reference style in text, for example, Wander et al., 2011).

 Response: Thank you for the comments. The changes effected as suggested.

  1. Please provide a reference for SPSS

 Response: Changes effected as suggested. See line 152

Results

  1. Demographic characteristics of the respondents should be provided.

 Response: Changes effected as suggested.

  1. Please rephrase following sentence

Girls show a higher significant (P<0.05) mean SBP and DBP than boys. Mean SBP girls (104.257mmHg) and boys (101.396); mean DBP girls (62.637) and boys (60.394).

 Response: Changes effected as suggested.

  1. Table 1, please switch a column and a row (like table 3) ; Please replace as * p<0.05; ** p<0.01

 Response: Thank you for the comments. Changes effected as suggested.

  1. Table 2. I would suggest to switch the column BP <90th percentile and the column BP>90th percentile for better readability

 Response: Thank you for the comments. Changes effected as suggested. Table 1 included both the prevalence of undernutrition, mean and standard deviation of the variables of interest.

  1. Figure 1. Please remove the title shown in the figure

 Response: Figure 1 has been deleted and be replaced by comparison of mean values of Ellisras children with the NHANES III reference population.

  1. 5. Please pay attention to numbers in paragraph 1 There is, however, a weak positive significant (p<0.05) correlation between TUAA, UMA, and BMI in both boys and girls (0.172**, 2.68**, 0.200**, 0.286**, 0.152** and 0.0181**).

Response: The sentence has been deleted.

  1. There are no explanations of negative correlations.

 Response: Thank you for the comments. we have left out the correlation tables  and hope the regression analysis could answer the question of association better as it account for confounder of age and gender. Secondly, we have let it as we wanted to keep the tables and the figures to be seven  as it is a standard peer review Journal practice. However, we have argued in the discussion that undernutrition could lead to cardiovascular diseases later in life. See line 275 to 283.

  1. Overall results section need to be revised in order to meet guidelines of the IJERPH.

 Response: Changes effected as suggested.

Discussion

  1. 6. Pay attention to reference style.

 Response: Changes effected as suggested.

  1. There are no discussion in the discussion section. It seems like abstract plus limitation of the study. Please provide discussions in depth

 Response: Changes effected as suggested.

Round 2

Reviewer 1 Report

The author has answered to all my requirements.

Author Response

Review 1:

The author has answered to all my requirements.

Response: Thank you. Much appreciated

Reviewer 2 Report

The reviewer acknowledges that the authors’ addressed all of the reviewer’s feedback, as well as included new results to the manuscript. However, with the addition of new results the reviewer has additional comments that needs addressing, as well as further comments to improve the clarity and readability of the manuscript.

Some comments are included below and others are provided on the electronic manuscript that refer to minor grammatical changes.

Introduction.

Further clarification of what is currently known and what is not known regarding body composition measures and cardiovascular disease/hypertension risk in children should be included.

It is suggested to include reference to it is known that childhood obesity is associated with increased blood pressure although the potential cardiovascular risk later in adulthood of undernourished children is less clear. Interweaving the current evidence provided in the introduction to improve linking and clarity on the significance of the study.

Context needs to be provided to why the study compare the findings to the NHANES III study, is this a comparable group? Why is this group chosen as a reference group?

Conclusion.

This section can improve to include the main takeaways of the importance of the study findings within the existing body of knowledge as well as future directions.

Author Response

Review 2

Introduction.

Further clarification of what is currently known and what is not known regarding body composition measures and cardiovascular disease/hypertension risk in children should be included.

Response: Thank you for the comments. Corrections effected as suggested. See line 49 -59.

It is suggested to include reference to it is known that childhood obesity is associated with increased blood pressure although the potential cardiovascular risk later in adulthood of undernourished children is less clear. Interweaving the current evidence provided in the introduction to improve linking and clarity on the significance of the study.

Response: Thank you for the comments. The correction effected as suggested. See line 49-59

Context needs to be provided to why the study compare the findings to the NHANES III study, is this a comparable group? Why is this group chosen as a reference group?

Response: The comments effected as suggested. See line 49-59.

Conclusion.

This section can improve to include the main takeaways of the importance of the study findings within the existing body of knowledge as well as future directions.

Response: The comments effected as suggested. See line 259-262

Comments on the manuscript.

Response: Comments effected as suggested. However, we could not test significant difference between NHANES population sample and the children in the current study as the reviewer has suggested. Furthermore, the sentence on Assent form have been deleted.

Reviewer 3 Report

This manuscript has been improved significanlty. however my comments on correlation should be mentioned properly (I don't find argument on this matter as  author mentioned).

Author Response

Review 3

This manuscript has been improved significantly. however my comments on correlation should be mentioned properly (I don't find argument on this matter as  author mentioned).

Respond: The two previous comments and respond on correlation were:

  1. a). Pg. 5. Please pay attention to numbers in paragraph 1 There is, however, a weak positive significant (p<0.05) correlation between TUAA, UMA, and BMI in both boys and girls (0.172**, 2.68**, 0.200**, 0.286**, 0.152** and 0.0181**).

Response: The sentence has been deleted.

  1. There are no explanations of negative correlations.

 Response: Thank you for the comments. we have left out the correlation tables  and hope the regression analysis could answer the question of association better as it account for confounder of age and gender. Secondly, we have let it as we wanted to keep the tables and the figures to be seven  as it is a standard peer review Journal practice. However, we have argued in the discussion that undernutrition could lead to cardiovascular diseases later in life. See line 275 to 283.

Response: Thank you for the comments.

  1. The correlation values were low. We have remove the correlation table from the study as we could not adjust for age and gender in the analysis. However, the linear regression model in the current study also show weak association but the association is significant (P<0.000) even after adjusting for age and gender. See line 189 -195 and line 217 to 226